# Evidence of stage progression in a novel, validated fluorescence-navigated and microsurgical-assisted secondary lymphedema rodent model

P. A. Will[1]*, A. Rafiei[1], M. Pretze[2], E. Gazyakan[1], B. Ziegler[1], U. Kneser[1], H. Engel[1,3], B. Wängler[2], J. Kzhyshkowska[4,5], C. Hirche[1]

1 Department of Hand, Plastic, and Reconstructive Surgery, Microsurgery, Burn Centre, BG-Trauma Hospital Ludwigshafen, Ludwigshafen, Germany, 2 Department of Clinical Radiology and Nuclear Medicine, Medical Faculty Mannheim, University of Heidelberg, Mannheim, Germany, 3 Ethianum Klinik Heidelberg, Heidelberg, Germany, 4 Institute of Transfusion Medicine and Immunology, Medical Faculty Mannheim, University of Heidelberg, Mannheim, Germany, 5 German Red Cross Blood Service Baden-Württemberg—Hessen, Frankfurt, Germany

* Patrick.will-marks@bgu-ludwigshafen.de

**Data Availability Statement:** All relevant data are within the paper.

## Abstract

Secondary lymphedema (SL)is a frequent and devastating complication of modern oncological therapy and filarial infections. A lack of a reliable preclinical model to investigate the underlying mechanism of clinical stage progression has limited the development of new therapeutic strategies. Current first line treatment has shown to be merely symptomatic and relies on lifetime use of compression garments and decongestive physiotherapy. In this study, we present the development of a secondary lymphedema model in 35 rats using pre- and intraoperative fluorescence-guided mapping of the lymphatics and microsurgical induction. In contrast to the few models reported so far, we decided to avoid the use of radiation for lymphedema induction. It turned out, that the model is nearly free of complications and capable of generating a statistically significant limb volume increase by water displacement measurements, sustained for at least 48 days. A translational, accurate lymphatic dysfunction was visualized by a novel VIS-NIR X-ray ICG-Clearance-Capacity imaging technology. For the first-time SL stage progression was validated by characteristic histological alterations, such as subdermal mast cell infiltration, adipose tissue deposition, and fibrosis by increased skin collagen content. Immunofluorescence confocal microscopy analysis suggested that stage progression is related to the presence of a characteristic $\alpha$ SMA⁺/HSP-47⁺/vimentin⁺ fibroblast subpopulation phenotype. These findings demonstrate that the *in-vivo* model is a reliable and clinically relevant SL model for the development of further secondary lymphedema therapeutic strategies and the analysis of the veiled molecular mechanisms of lymphatic dysfunction.

**Funding:** The authors received no specific funding for this work.

**Competing interests:** There are no conflict of interest or competing interest to disclose by the authors.

## Introduction

Secondary lymphedema (SL) is a significant complication after oncological therapy and filarial infections. It is associated with disfiguring appearance and psychological morbidity [1]. SL is one of the leading disability causes, affecting as many as 140 to 250 million people worldwide, associated with a huge burden to the healthcare system [2–4].

In western populations, breast cancer is not only the most frequent female cancer but also the main cause of SL [5]. As a late complication of breast cancer therapy, 5–70% of the patients suffer from SL depending on the extent of surgical lymph basin dissection and adjuvant onco-logical therapy [6]. Furthermore, SL occurs commonly as a surgical complication in skin (28%), gynecological (20%), and urological (10%) cancers [7–9].

SL becomes manifest when extravasated fluid remains accumulated in the intercellular space followed by locoregional soft tissue alterations. If this phenomenon endures, a chronic form of hardened swelling, fibrosis, adipose tissue accumulation, immune cell infiltration, and limb deformation occurs [7,10,11]. This collective of progressive and sequential morphological changes are defined as stage progression [12].

Current treatment strategies consist of a multimodal approach, including exercise, skin care, compression bandaging, and complex decongestive physiotherapy [13], as the leading therapeutic strategies are merely symptomatic. In recent years, causal therapies like lymph node transfer (LNT) [14] and lymphovenous anastomosis (LVA) [15] have emerged. These supermicrosurgical procedures require sophisticated surgical equipment and exceptionally skilled microsurgeons that are not available in most developing countries [16]. Furthermore, supermicrosurgical procedures have proven limited effectiveness [17] and when LNT is per-formed, an iatrogenic donor site SL might develop [18]. Currently, drugs or cellular therapy do not play a therapeutic role, since the effects of cellular pathogenesis, immunological regula-tory mechanisms, and the molecular key players of stage progression have only been identified to a limited degree [5].

To understand the molecular mechanisms causing this disease and to further investigate the pathophysiology of stage progression, a simple, yet reproducible and validated animal model would be useful. Despite the fact that several rodent tail models for lymphedema have been reported [19–23], none employed a selective lymphatic excision but an indiscriminate dermal incision of practically the entire circumference of the tail, overlooking that damage of the venous plexus and main dermal veins might be the cause for the observed edema by means of venous insufficiency. Histologically, most of the rat tail is epidermis and cartilage [24], con-sequently, rodent tail lymphedema models might resemble human limb lymphedema inaccu-rately from a translational an anatomical perspective. Therefore, some rodent limb models for SL were proposed (Table 1).

Unfortunately, the currently published models for SL fail to induce a solid and persistent SL [25] presented several major translational drawbacks (Table 1). Some key pitfalls to overcome are unreliable *in-vivo* generation of SL, high complication rates, missing diagnostic criteria for induced SL, and absence of translational disease progression hallmarks [25]. Hydrophilic blue dyes, applied by all previous studies to identify the lymphatics, are also taken-up by connective tissue and veins (example shown in Fig 1I). An unprecise excision could trigger non-lymph-edema induced fibrosis or limb swelling following vein insufficiency [25].

Near-infrared imaging of indocyanine green (ICG) is a standard procedure in the clinical assessment of the lymphatic system [26–29]. ICG based NIR imaging has been used both to evaluate lymphatic function in animal models and NIR functional imaging [20,29–31], never-theless, it has not been used as a standardized tool for pre- and intraoperative lymphangio-graphic navigation to guide a precise lymphatic injury for *in-vivo* studies.

**Table 1. Descriptive summary of the methodology and pitfalls of the existing rat Secondary Lymphedema (SL) models.**

| Rat hind limb model | Methods for LE induction | N | Clinical outcome | Definition of LE | Functional outcome | Histological outcome analysis | Comments |
|---|---|---|---|---|---|---|---|
| Wang et al. (1985) | Skin strip and popliteal LN and LV excision. Tracer: 0.5% azo-blue | 70 | Mean circumferential and volumetric water displacement measurement | **Not reported** | **Not reported** | H&E staining of paraffin sections with histological analysis | Mortality rate of 21% |
| Kanter et al. (1990) | 6 groups of different combination of femoral vein obliteration, skin strip resection and 4.5 Gy RT. Tracer: 0.05ml isosulfan blue | 28 | Mean circumferential and volumetric water displacement measurement | **Not reported** | Tc-99 lymphoscintigraphy | **Not reported** | Mortality rate 8% |
| Huang et al. (1990) | Incision of all soft tissue (including muscles) circumferentially with no skin, LN or LV resection. Tracer: 0.1% methylene blue | 50 | Mean circumferential measurement | **Not reported** | **Not reported** | H&E staining of paraffin sections with histological analysis | Mortality 4% |
| Lee-Donaldson et al. (1999) | 4 groups of skin strip and femoral LN and LV excision and RT with 4.5 Gy. Tracer: 1% Evans blue | 45 | Mean circumferential measurement; Subgroup indirect by MR imaging (n = 3) | **Not reported** | Tc-99 lymphoscintigraphy (n = 9) | **Not reported** | Mortality up to 13%, skin breakdown and, bone necrosis |
| Kawahira et al. (1999) | Mice bone marrow MF isolated by FACS. Patent blue | 26 | Mean circumferential measurement | **Not reported** | **Not reported** | H&E staining of paraffin sections with histological analysis | No real model, more a primordial LNT |
| Yang et al. (2014) | Whole inguinal fat pad and popliteal LN removal. Dyed vessel removal. No Skin gap left. Radiation 20–40 Gy Tracer: 0.1ml Evans blue | 71 | Indirect by CT imaging and volumetric calculation | **Not reported** | Tc-99 nanocolloid -SPECT lymphoscintigraphy | **Not reported** | Mortality and complication rate high (38%) Low success rate (57.7%) |

**Abbreviations:** LE = Lymphedema LN = Lymph node RT = Radiotherapy LNT = Lymph node transfer LV = Lymph vessel MR = Magnetic resonance.

Notorious volumetric regression was reported in several SL models, as a consequence, many authors added high dose radiotherapy to the surgery in order to generate an SL [25]. Radiation might account for the high complication rates in preexistent SL models. Even more, a differentiation between the outcomes of the subsequent actinic process and SL is not feasible. Further, the underlying molecular disease mechanisms of SL stage progression would be impossible to discriminate from the radiation-induced chronic inflammation.

Stage progression is a key feature of the disease but, upon now, has only received limited attention in the current literature [12], and has not been shown in a rat model as a consistent process over the follow-up period. Isolated pathophysiological components of stage progression like H&E dermal alterations [29,32], increased adipose tissue [33,34], mast cell infiltration [35,36], and sclerosis [32,33,37,38] have been reported. Furthermore, a decisive methodological aspect is missing in all the preceding *in-vivo* SL models: a translational definition of SL [25]. Reported induction rates are meaningless when diagnostic criteria are lacking. To overcome these important challenges and restrictions, our objective was to establish a reproducible *in-vivo* model for SL with clear definitions and translational outcome assessment, by quantitative and standardized clinical, functional, and morphological analysis.

## Materials and methods

### Development of rodent microsurgical lymphedema model

Female albino Lewis rats (n = 35; Lew/Crl-RT1 each 250–270 g) were purchased from Charles-River Laboratories (Sulzfeld, Germany). All animal procedures were performed according to

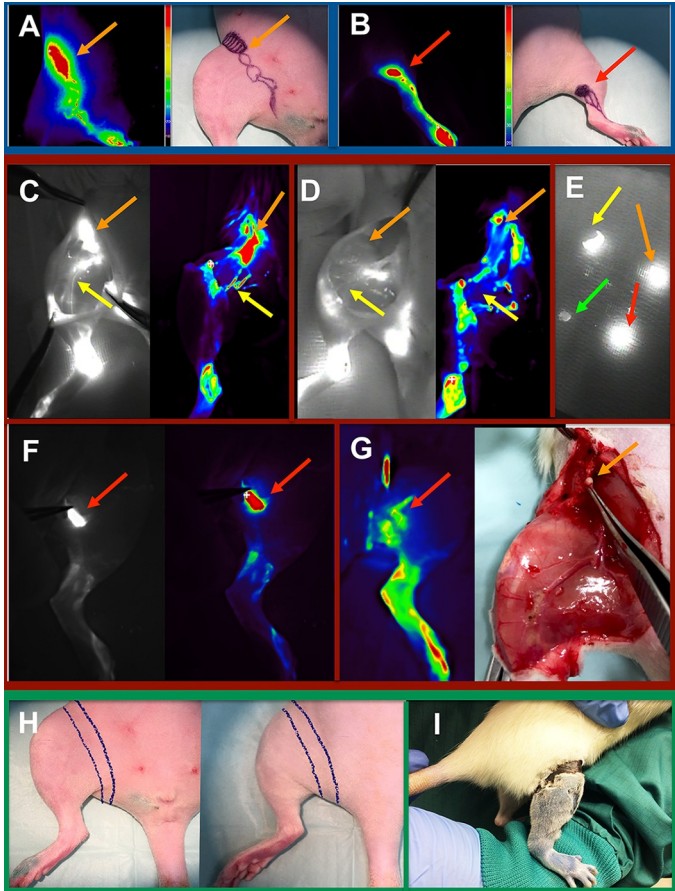

**Fig 1.** **(Blue box)** **Preoperative** *in-vivo* **near-infrared, fluorescence-guided mapping with Indocyanine green (ICG) and Fluobeam® camera.** A color-coded mapping of the lymphatic system of the inguinal **(A)** and popliteal **(B)** regions guided the preoperative markings. **(Red Box)** **Intraoperative** *in-vivo* **near-infrared, fluorescence-guided mapping and navigation including color-coded visualization.** In **(C)** the inguinal lymph nodes (orange arrows) and lymphatic vessels (yellow arrows) are identified before excision. Fluorescence-guided intraoperative control of the removed inguinal structures is showed in **(D)**. The red arrow indicates intraoperative vision of a popliteal lymph node before **(F)** and after microsurgical excision **(G)**. Intraoperative ICG navigation permits highly sensitive fluorescence differentiation of tissues **(E)**. Orange and red arrows show removed inguinal and popliteal lymph nodes respectively, while the yellow arrow shows lymphatic vessels in fascia and the green arrow resected vein tissue. **(Green Box)** In **(H)** the skin strip to be circumferentially excised is marked intraoperatively. The postoperative skin gap after suturing is shown in **(I)**.

European animal welfare guidelines and regulations, after gathering the approval of the local ethic committee (BG Klinik Ludwigshafen) federal authorities. For this project, the specific approval of the states of Rheinland-Pfalz (Landesuntersuchungsamt Mainz, Nr. G-206-15) and Baden-Württemberg (Regierungspräsidium Karlsruhe, Nr. 35–9185.81) were obtained prior to animal experimentation.

The right hind limb served for intervention, while the left hind limb served as a control (untreated). Isoflurane (2–4%) was used for anesthesia. Following 0.01 mL subcutaneous injection of 5 mg/mL indocyanine green (ICG) solution (Verdye®, Diagnostic Green, Germany) on the dorsum of the shaved right paw and the inner thigh 2 cm lateral to the vagina, the main lymphatic vessels and lymph nodes were identified with a highly sensitive near-infrared camera (NIRC) (Fluobeam 800 NIR, Grenoble, France) after a latency of 5min for dye uptake. The lymphosomes [39] were mapped on the right ventral and dorsal limb and marked on the

shaved skin prior to incision (Fig 1A and 1B). Next, a vertical oriented incision was made in the mid-groin, and an 8–10 mm wide strip of skin and subcutaneous tissue was excised circumferentially down to the fascia (Fig 1H and 1I). The superficial lymphatics were surgically removed within the skin strip as previously reported [40]. NIRC was fixed at 20 cm above the operating site, and the whole intervention was visualized under real-time navigation of the lymphatic structures (Fig 1C–1G).

Using a microsurgical microscope for magnification (Carl Zeiss F170, Jena, Germany) and microsurgical instruments, the subcutaneous tissue was dissected, and the deep lymphatics and lymph nodes skeletonized carefully, without injuring blood vessels. Then, the deep lymphatic vessels were excised in addition to the popliteal and inguinal lymph nodes. To prevent reconnection of the superficial lymphatics, the wound edges were sutured with Poly-p-dioxanon 5–0 (PDS) (Ethicon, Norderstedt, Germany) to the groin muscle fascia, and a gap of 5 mm was left (Fig 1I). Wound closure spray (Opsite®, S&N, Hamburg, Germany) was applied as a wound dressing after wound closure.

Postsurgical care consisted of isolation of the animals for 2 days and subcutaneous application of weight adapted (0.05 mg/kg) buprenorphine three times per day for the first five days. The medication was adapted based on distress monitoring 5 times per day with the rat grimace scale protocol. Physiological circadian cycles of rodents were respected by automated and preprogrammed lights. Enrichment materials as well as shelter were permanently provided. After 48 h, the isolated rats were returned to the original cage in order to respect normal social interactions. Prior and after interventions, rats were kept with food and water *ad libidum*. No perioperative antibiotic treatment, NSAIDs or steroids were used. At a predetermined timepoint the rats were sacrificed by Narcoren® i.v application and cervical dislocation under deep Isoflurane anesthesia.

## Objective validation of induced secondary lymphedema

Postoperative complications were monitored and documented. For SL diagnosis, the clinical equivalent human diagnostic criteria (increase of limb circumference >2 cm, volume more than 200 cm³ or >10% of volume increase compared to contralateral) were applied translationally to the rats [41]. For this study, an increase of the limb circumference of >10% was set as the threshold for clinical diagnostic of SL. Water displacement method was used during this study for the volumetric assessment. While muscle relaxation was achieved under anesthesia, each hind limb was completely introduced up to the inguinal region into a 58 mL polypropylene tube, filled with an exact volume of 58 mL of water. While introduced, the limb displaced water proportional to its volume. The remaining water was measured, and the hind limb volume calculated.

This procedure was repeated independently three times at the day of surgery and at postoperative days (POD) 1, 3, 5, 7, 10, 14, 21, 30, and 45. The mean values were documented. In order to overcome a potential measurement bias, such as subconsciously dipping the operated limb further inside the tube, the polypropylene tube employed were fitted to specifically block the limb submersion once the inguinal region was reached. To avoid inter-operator variability, all measurements were performed by the second author. Since the majority of reported incidences of SL were described in the first 3 years after ceasing the oncological therapy in patients, the follow-up period was set as a four human year equivalent. Rats older than 63 days have a translational equivalent of one human year every 11.8 days [42], representing a translational follow-up period of 45 days after the SL induction.

## Physiological analysis of the lymphatic system

To compare the lymphatic dysfunction and dermal backflow pattern, we injected the dorsum of the paw and inner upper thigh with each 0.01 mL ICG s.c. at POD 1, 3, 5, 7, 10, 14, 21, 30, and 45 under isoflurane anesthesia. After the ICG application on both limbs, rats were positioned inside a multispectral bioluminescence VIS-NIR X-ray imaging optical device (In vivo Xtreme, Bruker, Ettlingen, Germany), one hour after the injection. At that time point, simultaneously a whole-body X-ray, optical image, and a highly sensitive bioluminescence image were acquired. The settings were: Exposure time 30 sec, field of view 18 cm, X-ray filter 0,8 mm, 45 kVP 10 sec, reflectance 1 sec, fluorescent excitation wavelength 750 ± 20 nm, and emission wavelength 830 ±20 nm. By this modality, the lymphatic anatomy was contrasted and detected by means of the fluorescence signal, which was overlaid with bone and soft tissue anatomical landmarks provided by the X-ray and optical image. The ICG VIS-NIR X-ray imaging was always performed at 1h after injection because it was observed that the ICG signal intensity on intact limbs decreased progressively until the first hour at the pilot group while establishing the protocol. After that timepoint only a NIR imaging signal was detected at the injection sites (S1 Fig).

Using the VIS-NIR settings described above, the measurements were obtained from the intervened limb and the control simultaneously. The capacity of transporting the ICG from the peripheric tissue and clear it into the venous system was called ICG clearance capacity (ICC). The results were obtained by two approaches. First, semi quantitatively by bioluminescence signal overlaid too soft tissue with a color correlation of the signal intensity to a scale bar. This pictures were later classified into Yamamoto dermal backflow pattern [43] by a blinded observer. Second, at the X-ray images overlaid with the ICG fluorescence signal a region of interest (ROI) was selected at the intervened and control extremities (excluding the injection sites). Here, the software of the VIS-NIR X-ray imaging optical device determined the median ICG fluorescence intensity as counts per second (cps). Then the mean background cps of the image was subtracted from the median fluorescence value of each measured limb in order to get semiquantitative data of the ICC fluorescence signal.

## Morphological stage progression and fibroblast phenotyping

As a translational aspect of chronic SL, we defined stage progression in this project as histopathological dermal changes, local cellular alterations, and modification of the connective tissue, consecutive to interstitial progressive fluid accumulation over time. To evaluate this and after clinical and physiological measurements, the rats were sacrificed at POD 1, 3, 5, 7, 14, 21, 30, or 45. At these specific time points, skin tissue samples from the foot (zone A), popliteal region (zone B), and thigh (zone C) were obtained of the intervened limb (right) and control limb (left). The samples were harvested at a size of 0.5 x 0.5 mm, fixed in 4% PFA, and paraffin embedded or snap frozen in 2-metylbutan. For the staining, 5 μm tissue samples were cut out of the sample blocks (Leica CM3050S, Wetzlar, Germany). In the paraffin slides antigen retrieval was performed for 15 min in an alkaline heat-induced epitope retrieval solution (ZUC040, Zytomed, Berlin, Germany) at 95 ºC. To determine changes in the morphology and composition of the skin layers, H&E (Merck 109249, Darmstadt, Germany) staining was performed according the manufacture's protocol.

As recently published [44], collagen density at tissue sections can be detected and quantified by rhodamine fluorescence microscopy analysis after performing a pico-sirus red (PSR) staining. For detection of collagen fibers type 1 and 3, representing fibrosis (e.g. In SL), PSR (Sigma 365548, St Louis, USA) staining was performed. In order to get a quantitative approximation of the collagen content for the whole sample, each PSR stained slide was automatically scanned

with the MosaiX fluorescence module in Rhodamine and FITC filter mode (Zeiss Primo Star Microscope, Oberkochen, Germany). The results of the fluorescence analysis were converted into 32 bit grayscale Tiff images and with the use of ImageJ (Version 1.5i, U.S National Institutes of Health, Bethesda, USA) and saved in the hard drive.

The mean spectral overlap of the FITC channel, autofluorescence of cells and PEGs, was subtracted digitally from the Rhodamine signal to gain the mean intensity of the collagen fibers. In order to determine the mean collagen density, the mean intensity of the collagen fibers was normalized to the respective sample size (measured area in $mm^3$). Fat deposition, as a histological hallmark of late phases of stage progression in SL, was detected by Oil Red O staining solution (Sigma O1391, St. Louis, USA) in frozen slides. Toluidine blue (Sigma 89640, St. Louis, USA) was used in order to identify mast cells in the deparaffinized samples as a sign of chronic inflammatory changes.

To identify the presence of activated fibroblasts and their relationship with a profibrotic phenotypical pathway, double immunofluorescence (IF) staining of vimentin, α-SMA, collagen I, and heat-shock protein 47 (HSP-47) was performed. To identify co-expression confocal microscopy was used at 63-magnification (Leica TCS-SP8, Wetzlar, Germany). At predefined slide regions, the already obtained confocal images were randomly mixed in a Power Point presentation and the α-SMA⁺/HSP-47⁺/vimentin⁺ cells counted by a blinded observer. The mean count of triple positive cells per field of view for the different time points and zones was then compared and statistically analyzed.

## Statistical analysis

All statistical tests were performed using GraphPad Version 8.00 (La Jolla, California, USA). For central tendency and data dispersion, the arithmetic mean and standard deviation were obtained, and grouped into datasets. The dataset of the plethysmography (water displacement method) was tested for normal distribution using the Shapiro-Wilk omnibus normality test (α 0.05). Once the data proved to be normally distributed, first one-way ANOVA was tested in order to determine the overall statistic difference of the two paired groups (intervened and control limb).

The results were corrected with Barlett's test. Further, multiple comparisons between the different timepoints were performed by 2-ways ANOVA and the assumption of equal variability tested with Geisser-Greenhouse correction (α 0.05 and 0.01). As post-hoc correction tests primarily Sidak was used for the multiple comparisons, yet Bonferoni and Benjamini, Krieger and Yekutieli tests were also calculated (α 0.05). The dataset for ICG clearance capacity did not pass the D'Angostino & Pearsons normality test nor the Shapiro-Wilk omnibus normality test with α 0.05. Therefore Kruskal-Wallis test with Friedman correction was applied (α 0.05 and 0.01).

Multiple comparisons of the different timepoints was later performed using the two-stage step-up method of Benjamini, Krieger and Yekutieli with α 0.05. The statistical analysis of the mean fluorescence intensity of PSR stained collagen was performed with one-way ANOVA and corrected with Barlett's test at α 0.05, since the dataset distribution proved to be normally distributed by the Shapiro-Wilk test (α 0.05). The mean count of α-SMA⁺/HSP-47⁺/vimentin⁺ cells per field of view of the SL samples analyzed by confocal microscopy did not passed the Shapiro-Wilk omnibus normality test with α 0.05 and was analyzed for statistical significance with Kruskal-Wallis test at α 0.05. For a better understanding, the statistical tests used were added to the respective figure legends. To all the figures the statistical marking were added as follows * if p≤0,05, ** if p≤0,01, and *** if p≤0,001.

An overview of the experimental setting for the outcome analysis is offered in Table 2.

**Table 2. Summary of the experimental design.**

| Parameter | N = | Postoperative days | Methodology |
|---|---|---|---|
| Volumetric analysis | 34 | 1, 3, 5, 7,10, 14, 21, 30, and 45 | Water displacement |
| Lymphatic dysfunction- imaging | 34 | 1, 3, 5, 7, 10, 14, 21, 30, and 45 | Dermal backflow pattern |
| Lymphatic dysfunction- clearance | 34 | 1, 3, 5, 7, 10, 14, 21, 30, and 45 | ICG clearance capacity (ICC) |
| Histomorphological pattern | 34 | 1, 3, 5, 7, 14, 21, 30, and 45 | H&E staining |
| Chronic inflammation | 34 | 1, 3, 5, 7, 14, 21, 30, and 45 | Toluidine Blue staining |
| Collagen content | 34 | 1, 3, 5, 7, 14, 21, 30, and 45 | Pico sirus red staining and Rhodamine filter fluorescence microscopical analysis |
| Fat tissue deposition | 30 | 1, 3, 5, 7, 14, 21, 30, and 45 | Oil red O staining |
| Fibroblast phenotype | 34 | 1, 3, 5, 7, 14, 21, 30, and 45 | Double immunofluorescence staining for vimentin, α-SMA, and heat shock protein 47 |

## Results

### Perioperative outcomes: The proposed SL model is reproducible and free of major complications

In 34 rats out of 35 rats, the microsurgical induction could be performed according to the planned experimental study (success rate 97.1%). In one rat the popliteal lymph node was missing. Representative images of intraoperative navigation and *in-vivo* mapping, before and after the intervention are shown in Fig 1.

No postoperative mortality was observed. In only five out of the 34 rats (14.7%), a self-biting behavior into the wound edges, was observed despite proper analgesia. This complication was attributed to sutures itching or dermatitis related to the occlusive wound spray. This complication was resolved by changing the sutures to Monocryl 4–0 (Ethicon, Norderstedt, Germany) and obviating the wound dressing. In contrast to other SL rodent models reported in literature, no infection, wound dehiscence, soft tissue necrosis, bone necrosis, seroma, or bleeding were observed during the follow-up period.

### An *in-vivo* microsurgical model induces a consistent secondary lymphedema

The mean volume difference of the intervened (red) and control (blue) hind limbs, for the intervention day are shown and presented graphically in Fig 2. At the intervened hind limb, a categoric difference of the mean circumferences (higher than >10%) was observed on each timepoint after the induction, thereby being specific for SL induction. This difference was statistically significant with α 0.05 and α 0.01 for all timepoints except day 0 (day of surgery). After a clear and progredient volume increase in the intervened hind limb, a progressive mean volume reduction was observed from day 21 until the end of the follow-up period, without spontaneous regeneration.

### The limb with induced secondary lymphedema presents significantly reduced ICG clearance capacity and characteristic dermal backflow pattern

In the intervened hind limb, an overall significantly reduced ICG clearance capacity (ICC) was observed after 1 hour of s.c. ICG injection when compared to the control limb (Fig 3). In the intervened right limbs, ICG seemed to accumulate particularly in the thigh and the gluteal region, in contrast, the left hind limbs presented complete, physiological, clearance. While during the first week a "splash" dermal pattern was observed at the right limbs, starting in the second week and until day 45, a "star dust" dermal backflow pattern was predominantly noticed. This pattern evolved into a diffuse dermal backflow pattern around day 45 and after (Fig 3). After day 21, a slight ICC normalization was detected in the flare pattern of the NIR-VIS

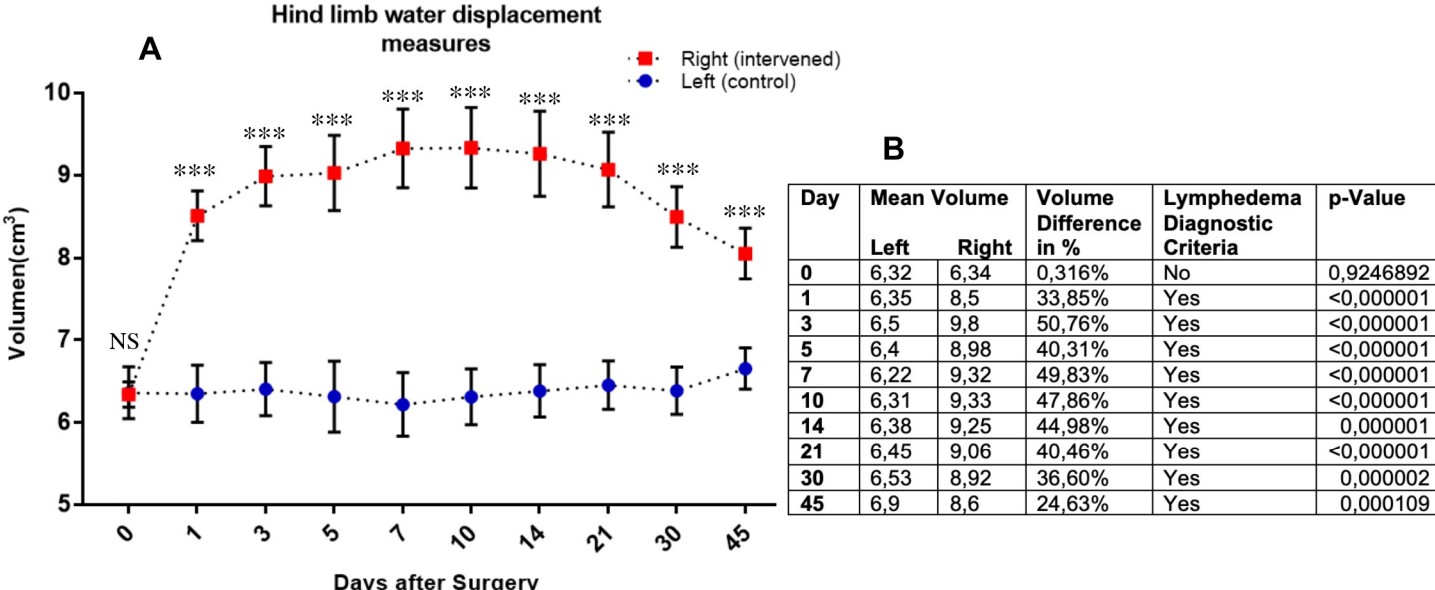

**Fig 2. Plethysmography by hind limb water displacement measures after induction of secondary lymphedema.** The mean volume differences (with SD) between the interventional (red) and control (blue) hind limb groups, related to the days after intervention, are displayed graphically in **(A)**. A numerical detail of the mean volume of the groups is provided in **(B)**. Statistics: One-way ANOVA for the overall statistic difference of the two paired groups and 2-ways ANOVA and Sidak post-hoc correction test. For the comparisons between the different timepoints.

analysis. The mean fluorescence intensity difference of the intervened and the control limbs (Fig 3) resulted statistically significant for all timepoints after the first postinterventional day with α 0.05 and α 0.01. Interestingly, the ICC showed a recovery after POD 14, with a lower mean cps intensity on POD 21, 30, and 45. In summary, despite an ICC recovery after POD 14, a clear lymphatic dysfunction was detected in the NIR-VIS analysis as part of model validation.

## Lymphedema stage progression: Dermal hypereosinophilia, skin layer distortion and sebaceous gland expansion are SL benchmarks in the rodent model

In the first 21 postoperative days no tissue architecture or staining intensity difference were observed in the 34 SL samples from zone A, B, or C. In contrast, at POD 30 all SL samples

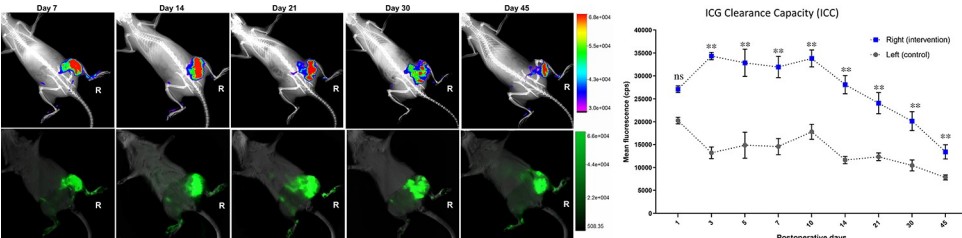

**Fig 3. Impaired lymphatic ICG clearance capacity (ICC) of the intervened limb is shown with color-coded mapping (upper row) and without color-coding (lower row) for POD 7, 14, 21, 30, and 45.** In the upper row of the figure the fluorescence intensity after 1 hour of injection is shown with flare mode merged with X-ray, for anatomical relation with the skeleton of the rat. In the lower row, ICG fluorescence is shown overlaid with the soft tissue to determine the dermal backflow pattern after 1 hour of injection. **Mean ICG clearance capacity (ICC) is presented on the right side.** The mean ICG clearance capacity (with SD) of the intervened (blue) and control (grey) hind limbs are shown for POD 1, 3, 5, 7, 10, 14, 21, 30, and 45. Statistics: Kruskal-Wallis test with Friedman correction and for multiple comparisons of the different timepoints two-stage step-up method of Benjamini, Krieger and Yekutieli.

presented a stronger eosin signal compared to the corresponding control. Particularly, the dermal and subdermal regions stained into intense red, implying a higher protein content in SL (Fig 4). In addition, after POD 30 an evident distortion in the histological architecture was observed with no clear interphase between the superficial and deeper dermal layers and a substantial sebaceous gland expansion.

## Lymphedema stage progression: Increased collagen presence is detected progressively in the hind limb with secondary lymphedema

As shown in Fig 5, in some SL samples a much higher collagen density was observed by PSR analysis under IF microscopy with Rhodamine filter. The mean intensity quantification, determined with Image J for each timepoints, is also shown graphically in Fig 5. A higher mean collagen intensity was detected on the PSR stained samples obtained from zone A, B, and C of the limb with SL. These results suggest a higher collagen I and III density, fiber size, or augmented production in the SL samples. The data showed to be statistically significant after day 14 for zone B (p = 0,033) and zone C (p = 0,032). Zone A showed a trend but failed to be globally significant (p = 0,085).

## Lymphedema stage progression: A progressive dermal fat deposition is observed after the first week

After performing Oil Red O staining and analysis of frozen sections obtained from 30 out of 34 *in-vivo* induced SL, a dermal fat deposition was observed in zone B and C consistently. In zone A, no trend was determined. Despite a volumetric increase observed during the first week in the hind limb of the model, no significant adipose tissue proliferation was detected (Fig 4). Between the first and second week, a fat cell deposition in the deep layer of the dermis was observed in the skin samples with lymphedema (Fig 4). Around week 4, the adipose tissue proliferation protracted to the superficial dermal layer and gained volume diffusely until the second month of follow-up (Fig 4). No staining difference was detected at an epidermal level on the intervened limb skin samples, when compared to control.

## Lymphedema stage progression: Sub-epidermal mast cell deposition increases notably after SL induction

After performing toluidine blue staining of 30 different samples of the rodent SL model, a discernible trend was observed in zone A, B, and C. At POD 1 only isolated mast cells at a subdermal level were detected, in contrast, after POD 14 a significant increase of perivascular and subdermal mast cell deposition was observed (Fig 6). The mast cell deposition showed no notorious trend change between POD 14 and POD 45. Around POD 45 a clear increase in toluidine positive cells were detected (Fig 6).

## Secondary lymphedema samples presented a subcutaneous α-SMA⁺/HSP-47⁺/Vimentin⁺ myofibroblast phenotype

Confocal microscopy analysis of SL derived samples from the rat model, demonstrated a high co-expression of α-SMA⁺/vimentin⁺ cells forming clusters in the dermis (Fig 7), as well as a similar co-localization pattern for HSP-47 with vimentin in all sections examined (Fig 7). Under close examination, α-SMA⁺/HSP-47⁺/vimentin⁺ cells showed a morphology and markers distribution expected for myofibroblasts (Fig 7). After POD 30, α-SMA⁺/HSP-47⁺/vimentin⁺ myofibroblasts were remarkably more ubiquitously distributed in the dermis, suggesting a proliferative pattern. Matching, the expression of several

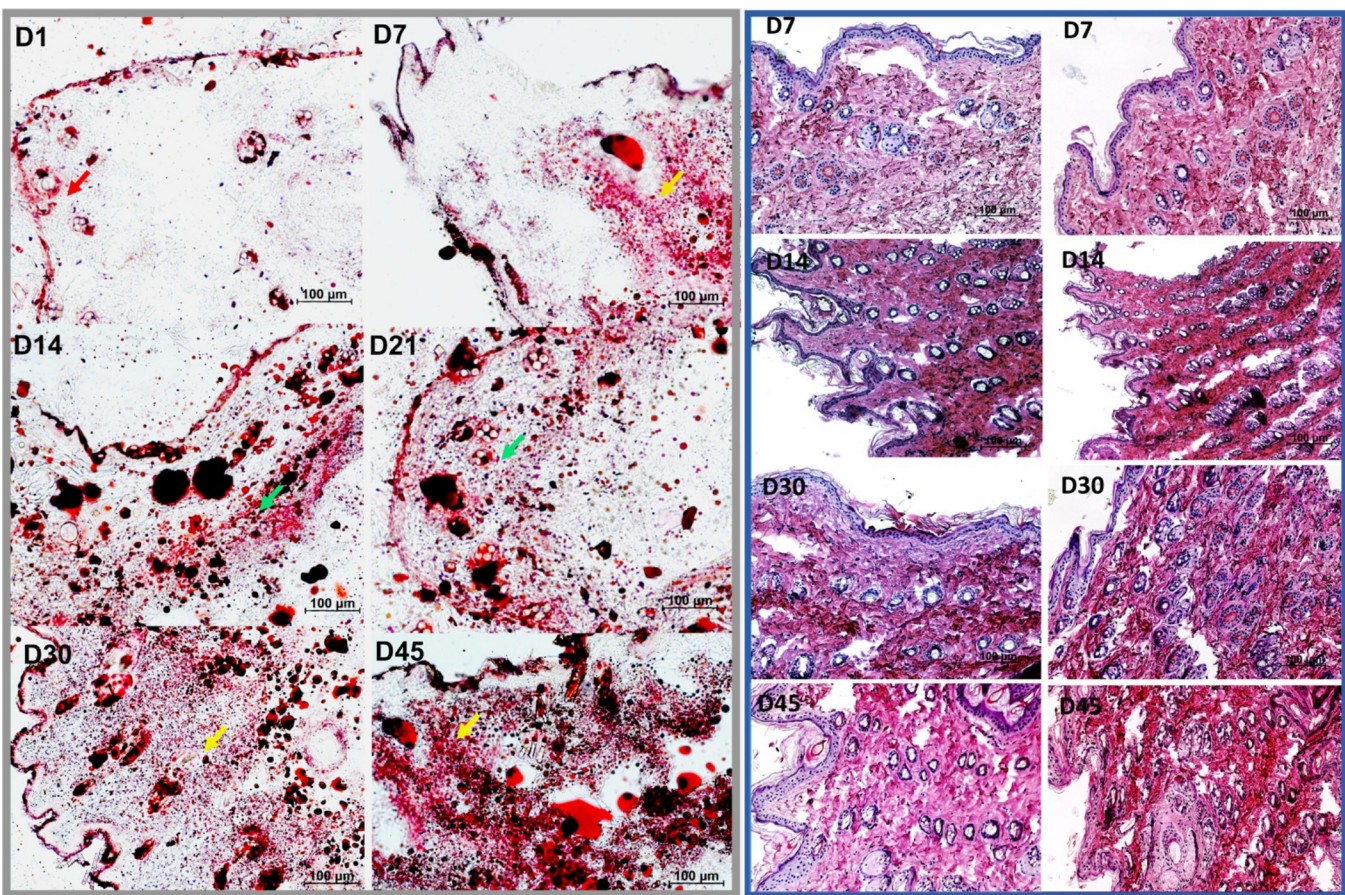

**Fig 4. Grey box: Microscopic analysis of Oil Red O staining of frozen tissue sections derived from skin samples of the secondary lymphedema *in-vivo* model.** Representative stained samples from different timepoints are shown on the left. After induction, only isolated lipids stained on the dermis (red arrow). After day 7, fat deposition in the deep dermis was observed (orange arrow). During postoperative D14 (week 2) and D30 (week 4) fat deposition was detected systematically in deep and superficial dermis (green arrows). After D30 a strong diffuse dermal fat deposition was detected (yellow arrows). Scale bar 100µm. Magnification 100x. **Blue box: Microscopic analysis of Hematoxylin & Eosin staining of paraffin sections derived from skin samples of the secondary lymphedema *in-vivo* model.** Representative stained samples from different timepoints are shown. On the left side, skin samples from the left limb serve as control. On the right, the samples of the right extremity, with secondary lymphedema, are offered as comparison. Scale bar 100µm. Magnification 100x.

strongly positive HSP-47 cells was observed in the dermis of the intervened limb after POD 30. As expected, HSP-47+ cells colocalized in ECM rich in collagen+ fibers at double immunofluorescence. The blinded analysis of triple positive cells per field of view revealed a general increase of α-SMA+/HSP-47+/vimentin+ cells in zones A, B, and C after day 14, reaching a maximum at POD 45. However, merely in zone C the mean counts of triple positive cells were found to be gradually increasing after the induction. On the other hand, in some of the samples at day 0 (control before induction) α-SMA+/HSP-47+/vimentin+ cells were detected. Nevertheless, the mean count per field was extremely low in the control slides, with mean counts of 0,53 cells for zone A, 0,26 cells for zone B, and 0,46 cells for zone C at day 0 (Fig 8). Despite the clear trend of increased counts of triple positive fibroblasts in lymphedema at later stages, zones A and B failed to be overall statistically significant.

In Fig 9 a graphic summary of the main results of the proposed rodent SL and the hereby generated evidence for lymphedema stage progression is offered.

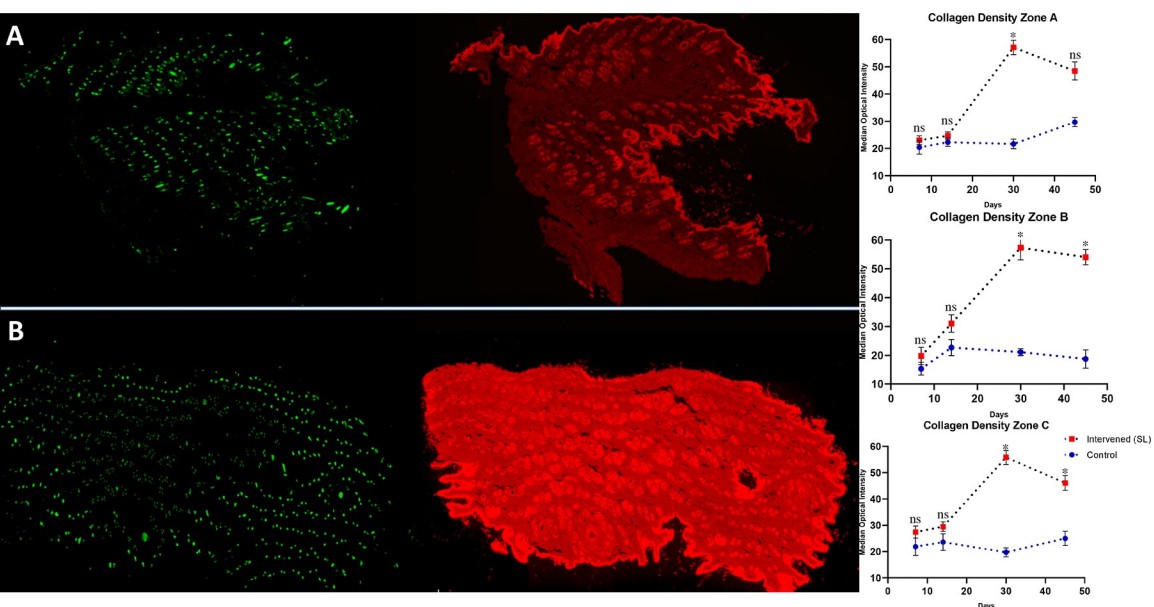

**Fig 5. Pico Sirus Red (PSR) fluorescence analysis of whole skin samples obtained from rat at day 30 after secondary lymphedema induction are presented at the left side. (A)** Whole skin section from zone B of the left, not intervened, hind limb is shown as a fluorescence scan. At **(B)** a representative whole skin section from zone B of the right, intervened, hind limb is presented as a fluorescence scan under FITC and Rhodamine filter. **On the right side, collagen density determined by PSR rhodamine fluorescence scan are shown for zones A, B, and C.** The median optical density of collagen fibers under Rhodamine filter fluorescence analysis is presented for days 7, 14, 30, and 45 in red for the intervened limb and in blue for the control extremity. Statistics: One-way ANOVA and correction with Barlett's test for the mean fluorescence intensity of PSR stained collagen.

## Conclusions and discussion

Translational research on SL addressing supermicrosurgical, immunological or cell-based therapies require a reproducible, validated, and stable rodent model. In contrast, the published models varied significantly in their methodology although shared significant limitations [25]. In the available hind limb models, radiation was used as a response to failed induction or spontaneous regression of SL [25]. To employ radiation during the induction of a SL *in-vivo* model might have the advantage to partially mimic the tissue transformations that occur in patients that underwent radiotherapy as part of their oncological management. On the other hand, the use of high radiation doses resulted in frequent and severe complications and might interfere in the natural development of SL stage progression due to unspecific inflammatory triggers and actinic fibrosis [25]. Further, mechanistic insights of the disease will be exceptionally difficult to investigate in a preexisting actinic microenvironment. To overcome these shortcomings, rodent tail models emerged as counterpart [20,23,45–47]. In such models a consistent edema was established without radiation and noteworthy molecular experimentations were performed [19,20,22,23,46,47]. Nevertheless, the rat tail in our opinion does not seem to anatomically, functionally, nor histologically be the optimal translational body segment compared to a human limb, which is mostly affected by SL. Furthermore, extremity models provide significant advantages in translational research as they involve the opportunity of a direct comparison to the unaffected extremity.

The lack of surgical refinement and intraoperative mapping of most rodent tail models are likely to induce unspecific soft tissue edema or venous insufficiency, in addition to a plausible lymphedema. Since in rodent tail models local lymph nodes are usually not excised, locoregional immunological and molecular findings might have an inadequate translational validity.

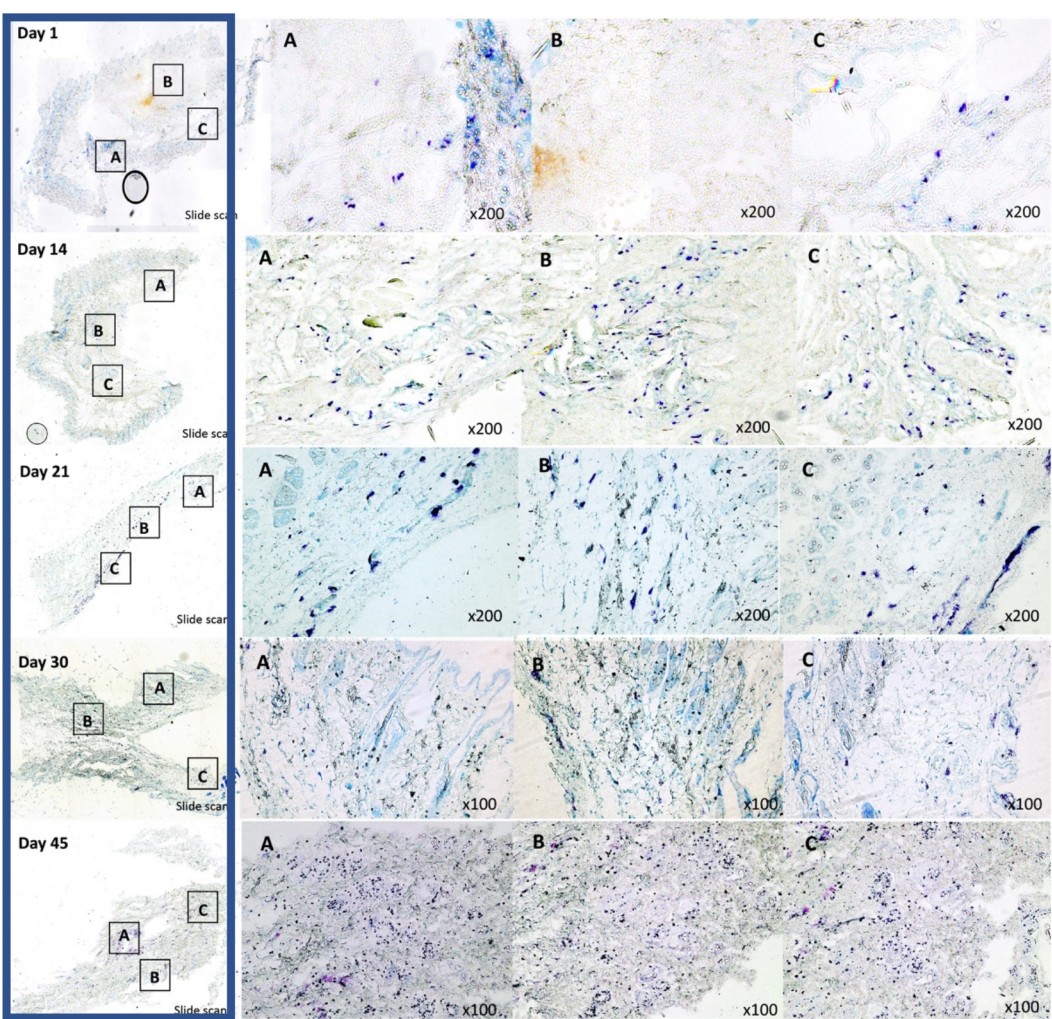

**Fig 6. Microscopic analysis of toluidine blue staining of paraffin embedded sections derived from skin samples of the secondary lymphedema *in-vivo* model.** In the blue box, the whole scanned slides for POD 7, 14, 21, 30, and 45 are offered as a visual reference. Closer magnification (x100 or x200) of each slide for representative areas of interest can be seen in respectively **A**, **B**, and **C**. If to many toluidine positive cells were observed in the area of interest, a x100 magnification was used to deliver a representative overview.

This fact is expressly relevant because lymphadenectomy is the most relevant contributing SL risk factor in oncological patients [8,9,41,48]. Besides the unclear overlapping of probable causes for the observed edema on the SL rodent hind limb and tail models, the current literature universally lacks a translational definition for SL, reporting hereby non-comparable success rates.

In the presented and validated *in-vivo* SL model with intraoperative fluorescence-guided ICG navigation and lymphosome mapping, unspecific soft-tissue and vein excision could be avoided. By real-time identification of lymphatic structures with ICG and the precision of microsurgery, the mortality rate of SL models could be reduced from 21% [25] to 0% in our study. Significant complication rates, reported up to 38% [49], were not observed in this model. The disparities might be explained by shorter surgery times, absence of neurovascular bundle lesions, and an effective induction of SL without the need for adjuvant radiation. During this project, for the first time, a clinically translated definition of lymphedema induction

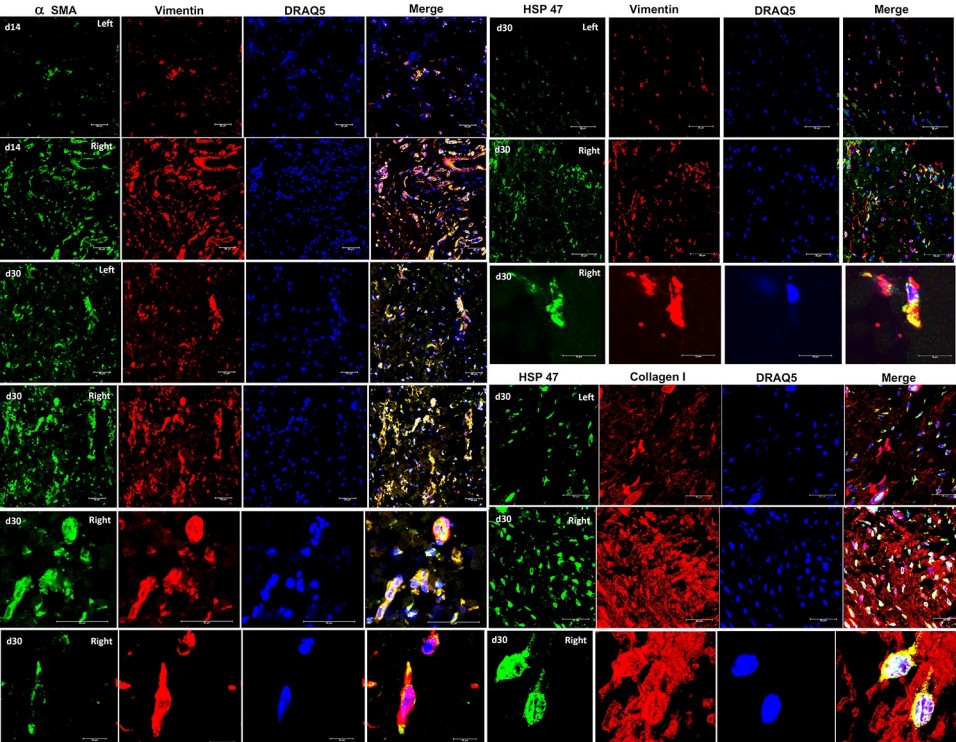

**Fig 7. Immunofluorescence co-expression of α smooth muscle actin (α-SMA), vimentin, collagen I, and heat shock protein 47 (HSP-47) in skin samples obtained from the rat SL model.** Left are control samples while right is intervened (SL) samples. The primary antibodies were mouse a- α-SMA or a-HSP-47 (visualized in green, Alexa flour 488 donkey a-mouse) and rabbit a-vimentin or rabbit a-collagen I (visualized in red, Cy3 goat a-rabbit). Nuclear staining was performed using DRAQ5 (visualized in blue). Co-localization in visualized in yellow in the merge channel. All scale bars are 50 μm while d30 indicates that the samples were obtained at day 30 after induction of secondary lymphedema.

was offered. A volumetric difference of 10% (translational definition for SL) in the intervened hind limbs after the induction, was so unmistakably achieved, that it was statistically significant even with p<0,01 for all the rats intervened, for the entire follow-up period, and further. The authors are aware that the chosen follow-up period in the rats, of 45 days, might not fully represent a translational equivalent of 3 human years since temporal kinetics of extracellular matrix, cell proliferation, enzymatic turnover rates, and DNA repair might vary interspecies during aging. In order to minimize this error the best available translational approximation currently available in the literature [42] was employed.

In the validation process of this model, the first step was to accurately detect lymphatic dysfunction. This could be achieved by the use of a simple, painless, fast, and radiation free technology. In our project we developed a protocol by VIS-NIR X-ray imaging technology for ICG-Clearance-Capacity, based on the principles of lymphangiography. A dermal backflow pattern, characteristic to SL chronification in patients [50], was shown for the first time in an animal model.

These clinical findings were validated by detection of abnormal ICG clearance capacity on the intervened hind limb, resembling patient's lymphatic dysfunction. Remarkably, the clinical and pathophysiological changes were so manifest that the SL *in-vivo* model achieved to be statistically significant with α 0.05 and α 0.01. Even after precise induction, clear SL development, lymphatic dysfunction, and histological stage progression, there was a trend to mean hind

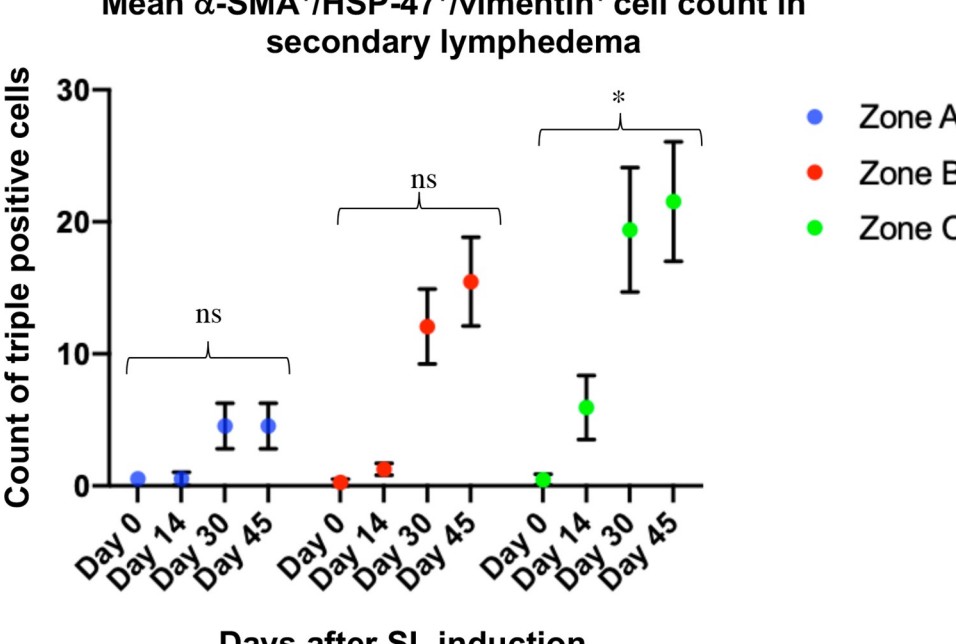

**Fig 8. Counts of α-SMA⁺/HSP-47⁺/vimentin⁺ fibroblasts in skin samples obtained from the rat SL model of zones A, B, and C at days 0, 14, 30, and 45.** The mean count of triple positive cells per field of view of all the samples at the different timepoints is shown in blue for zone A, in red for zone B, and green for zone C with the respective SD. The statistical analysis was performed with Kruskal-Wallis test.

limb volume reduction and partial recovery on the ICG clearance capacity on almost all animals after POD 21. This phenomenon, well known in scientific literature on rodents [25], was reported far more dramatically and earlier in appearance in the published models than what

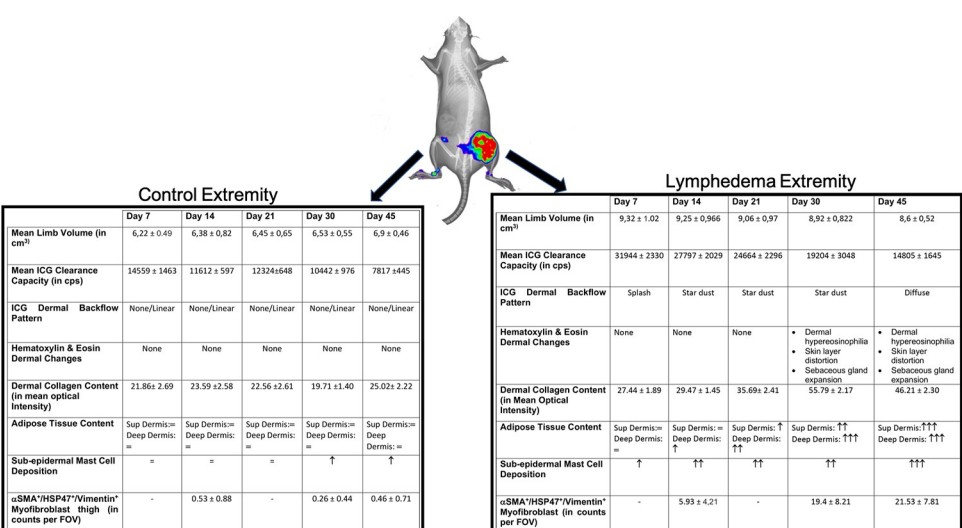

**Control Extremity**

| | Day 7 | Day 14 | Day 21 | Day 30 | Day 45 |
|---|---|---|---|---|---|
| Mean Limb Volume (in cm³) | 6,22 ± 0.49 | 6,38 ± 0,82 | 6,45 ± 0,65 | 6,53 ± 0,55 | 6,9 ± 0,46 |
| Mean ICG Clearance Capacity (in cps) | 14559 ± 1463 | 11612 ± 597 | 12324±648 | 10442 ± 976 | 7817 ±445 |
| ICG Dermal Backflow Pattern | None/Linear | None/Linear | None/Linear | None/Linear | None/Linear |
| Hematoxylin & Eosin Dermal Changes | None | None | None | None | None |
| Dermal Collagen Content (in mean optical Intensity) | 21.86± 2.69 | 23.59 ±2.58 | 22.56 ±2.61 | 19.71 ±1.40 | 25.02± 2.22 |
| Adipose Tissue Content | Sup Dermis:= Deep Dermis: = | Sup Dermis:= Deep Dermis: = | Sup Dermis:= Deep Dermis: = | Sup Dermis:= Deep Dermis: = | Sup Dermis:= Deep Dermis: = |
| Sub-epidermal Mast Cell Deposition | = | = | = | ↑ | ↑ |
| αSMA⁺/HSP47⁺/Vimentin⁺ Myofibroblast thigh (in counts per FOV) | - | 0.53 ± 0.88 | - | 0.26 ± 0.44 | 0.46 ± 0.71 |

**Lymphedema Extremity**

| | Day 7 | Day 14 | Day 21 | Day 30 | Day 45 |
|---|---|---|---|---|---|
| Mean Limb Volume (in cm³) | 9,32 ± 1.02 | 9,25 ± 0,966 | 9,06 ± 0,97 | 8,92 ± 0,822 | 8,6 ± 0,52 |
| Mean ICG Clearance Capacity (in cps) | 31944 ± 2330 | 27797 ± 2029 | 24664 ± 2296 | 19204 ± 3048 | 14805 ± 1645 |
| ICG Dermal Backflow Pattern | Splash | Star dust | Star dust | Star dust | Diffuse |
| Hematoxylin & Eosin Dermal Changes | None | None | None | • Dermal hypereosinophilia • Skin layer distortion • Sebaceous gland expansion | • Dermal hypereosinophilia • Skin layer distortion • Sebaceous gland expansion |
| Dermal Collagen Content (in Mean Optical Intensity) | 27.44 ± 1.89 | 29.47 ± 1.45 | 35.69± 2.41 | 55.79 ± 2.17 | 46.21 ± 2.30 |
| Adipose Tissue Content | Sup Dermis:= Deep Dermis: = | Sup Dermis: = Deep Dermis: ↑ | Sup Dermis: ↑ Deep Dermis: ↑↑ | Sup Dermis: ↑↑ Deep Dermis: ↑↑↑ | Sup Dermis:↑↑↑ Deep Dermis: ↑↑↑ |
| Sub-epidermal Mast Cell Deposition | ↑ | ↑↑ | ↑↑ | ↑↑ | ↑↑↑ |
| αSMA⁺/HSP47⁺/Vimentin⁺ Myofibroblast (in counts per FOV) | - | 5.93 ± 4,21 | - | 19.4 ± 8.21 | 21.53 ± 7.81 |

**Fig 9. Summary of the validation of the proposed SL rat hind limb model and the evidence generated for the lymphedema stage progression on the intervened extremity.** On the right, the clinical volume changes, ICG evaluation of the lymphatic insufficient, and the hystomorphological hallmarks of lymphedema stage progression are presented in a table. On the left the same outcomes are shown for the control extremities.

we observed in our model. However, and despite presenting stage progression, our rodent model is unlikely to provide an appropriately prolonged lymphedema to be ground for the study of elephantiasis-like skin modifications or oncological complications of lymphedema like Steward-Treves syndrome [51].

The mechanisms by which the rodent's regenerates function of the lymphatic system is still unknown, further investigations on the model might permit its comprehension and hereby the generation of new therapeutic concepts in lymphangiogenesis. It is fair to recognize that, none of the cited rodent models in this article, nor ours, might be suitable for the study of filariasis induced SL. The scientific community is currently certain, that in contrast to traumatic, surgical, or oncological caused lymphedema, the SL provoked by parasites are primary generated by autoimmune mechanisms in response to the infection [52,53].

In patients, after the onset of SL a progressive chronification occurs, called SL stage progression or migration [12]. This process is suspected to be regulated by an chronic inflammatory cascade, in which several pathophysiological mechanisms like fibrosis, adipose tissue deposition, and local immune cell infiltration coexist [38]. This is why a merely volumetric analysis for a translational SL *in-vivo* model seems inaccurate and insufficient. Despite the fact that numerous publications portrayed the central features of SL stage progression, until now a translational definition of SL was missing for *in-vivo* SL models. Consequently, stage progression as a progressive process with the co-occurrence of fibrosis, adipogenesis, histological modifications, and immune cell infiltration was not shown in SL animal models until now [25]. This crucial prerequisite for the development of translational therapies for SL was, to our consideration, not covered by the current *in-vivo* models.

In our model, despite the privation of radiation, the main components of chronic lymphedema were observed coexisting in a decisive histologic stage progression after POD 21. This was demonstrated by H&E hystomorphological modifications, categorical proliferation of sebaceous glands, adipose tissue deposition, mast cell infiltration, and dermal fibrosis by increased collagen type I and III content. A biphasic evolution of the induced SL was noticed. In the first 3 weeks (POD 21), a progressive increase in volume correlated with dysfunctional ICC but little or no histological changes. Such observations could be translated as the subacute phase of SL. This phase was followed by a slow, yet progressive, normalization tendency in the limb volume and improvement of the ICC after POD 21. However, during this period the permanent histological changes became evident. Such a bimodal evolution is not observed in patients, where the ICG transportation capacity and volume reduction remains pathologically or even worsens in correlation to histological alterations [5].

While in the literature the effect of the antigravitational postural changes of the human species, in contrast to rats, on the lymphatic network in normal and pathological conditions is still an unaddressed topic, several publications described spontaneously or postinterventional increased lymphangiogenesis in rodents [54–56]. In rats spontaneous VEGF-C mediated lymphangiogenesis was reported in chronic inflammatory states like renal fibrosis, ventricular remodeling, asthma, and peritoneal fibrosis [55,57–59]. Interestingly, VEGF-C seems to have a unclarified role in SL treatment and progression as it has been shown to improve [60,61] or worsen [62] SL. Fine-tuned regulation of the VEGF-C/VEGFR3 axis might elucidate the reason why rodents can revert the volume increase and lymphatic dysfunction in SL. Despite the regenerative potential of rats, the SL stage progression by means of histologic progression still happened. These initially contradictory findings might imply that the main pathophysiological mechanism of SL is not the chronic dysfunction on lymphatic transportation and interstitial fluid accumulation but a dysregulated molecular and cellular microenvironment.

Our data suggests that after reaching a threshold or a determined timepoint, and even after partial clinical resolution of the lymphedema and reconstitution of fluid drainage, some

undetermined locoregional factors accounts for the microenvironmental histological disbalance characteristic for a chronic inflammatory disease. Therefore, the most remarkable breakthrough in this study was the phenotyping of a specific version of tissue myofibroblasts as plausible explanation for local induction of lymphangiogenesis and stage progression co-occurrence.

Myofibroblasts are defined, in contrast to normal skin resident fibroblasts, by the expression of α-SMA and vimentin [63,64]. These cells undergo an epithelial to mesenchymal transition, gaining hereby a profibrotic phenotype and function [63–65]. After activation, myofibroblast are the major players in almost all main fibrotic diseases [66–69]. Myofibroblast-mediated fibrosis is generated by increasing collagen content, which in SL becomes clinically evident during stage progression [70]. HSP-47 is an indispensable chaperone in mammals for collagen formation and has been reported as an important biomarker for tumor angiogenesis [71], proliferation [72,73], and lymphatic metastasis [73] in several cancer cells and tumor associated fibroblasts. Upregulation of HSP-47 was also related to local wound healing impairment [74], organ fibrosis [75], and systemic inflammation in graft-versus-host disease [76]. Initial insights seem to link the pathogenesis of HSP-47 to TGFβ-1 pathways [77]. Using IF staining a co-expression of α-SMA$^+$/HSP-47$^+$/vimentin$^+$ cells was observed increasingly in all zones of the SL model corelated to stage progression. Especially in zone C (thigh) a statistical significance was observed in the mean count of α-SMA$^+$/HSP-47$^+$/vimentin$^+$ fibroblasts in contrast to zone A (food dorsum), were barely a small tendency in increased triple positive cells was detected at later SL stages. This result implies, that the site of subcutaneous injection of ICG (foot) is not likely to experience a tissue activation of profibrotic fibroblasts and that stage progression is unlikely to be affected by ICG imaging.

By a strong HSP-47$^+$ fibroblast subpopulation presence, particularly localized in samples with strong collagen content, a relation between infiltration of HSP-47$^+$ myofibroblasts and stage progression occurrence in SL can be postulated. In sum, HSP 47 myofibroblast phenotyping might be explored as an early diagnostic biomarker for SL stage progression. Further insight might reveal a TGF-β1 dependent activation of HSP-47 myofibroblast as therapeutic downstream target for the inhibition of stage progression [37]. In several cases an atypical cell morphology for fibroblast was observed. Those cells might have undergone macrophage to myofibroblast phenotype transition, as reported previously during chronic inflammation [65]. Consequently, the role of antigen presenting cells and molecular crosstalk to local myofibroblasts, as well as their immunomodulation potential during stadium migration, can be addressed in the *in-vivo* SL model with the certainty that the observations are independent from other sources of edema or radiation. Additional refinements in current SL models are crucial for the development of reliable translational supermicrosurgical, immunological, or cell based therapeutic strategies.

## Supporting information

**S1 Fig. Near-infrared images of the lymphatic clearance capacity in healthy rat limbs after subcutaneous application of ICG.** During preoperative mapping of the lymphatic system of the rat hind limbs and the intraoperative real-time navigation, an early appearance (2 min) of ICG contrast in the lymphatics was noticed by NIR visualization with the Fluobeam® camera. A clear contrast of the lymphatics of healthy limbs was detected with NIR visualization until 1 h after injection. By then most of the ICG was cleared from the limbs. However, at the subcutaneous injection sites, the ICG signal remained almost unaltered. This is probably to be related to the local binding of ICG molecules to tissue proteins. For further analysis, the injection sites were therefore not included in the region of interest (ROI).
(TIF)

## Acknowledgments

We want to acknowledge the logistical and technical contributions as well as advice for this study of Fabia Fricke, Matthias Schulte, Shanna Litau, Karen Bieback, Steffanie Uhlig, Alexander Marx, Juan Enrique Berner, Christina Schmuttermaier and Sayran Arif-Said.

## Author Contributions

**Conceptualization:** P. A. Will, E. Gazyakan, C. Hirche.

**Data curation:** P. A. Will, M. Pretze, C. Hirche.

**Formal analysis:** E. Gazyakan, U. Kneser, H. Engel.

**Investigation:** P. A. Will, A. Rafiei, M. Pretze, J. Kzhyshkowska.

**Methodology:** P. A. Will, A. Rafiei, U. Kneser, J. Kzhyshkowska, C. Hirche.

**Project administration:** P. A. Will, B. Ziegler, H. Engel.

**Resources:** M. Pretze, H. Engel, B. Wängler.

**Supervision:** U. Kneser, B. Wängler, J. Kzhyshkowska, C. Hirche.

**Validation:** A. Rafiei, M. Pretze, B. Ziegler, C. Hirche.

**Writing – original draft:** P. A. Will, J. Kzhyshkowska.

**Writing – review & editing:** P. A. Will, E. Gazyakan, B. Ziegler, B. Wängler, C. Hirche.

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
