## [Decision Letter · Decision Letter 0]

13 Feb 2020

PONE-D-20-00067

Evidence of stage migration in a novel, validated fluorescence-navigated and microsurgical-assisted secondary lymphedema rodent model

PLOS ONE

Dear Mr. Will,

Thank you for submitting your manuscript to PLOS ONE. After careful consideration, we feel that it has merit but does not fully meet PLOS ONE’s publication criteria as it currently stands. Therefore, we invite you to submit a revised version of the manuscript that addresses the points raised during the review process.

The manuscript is potentially interesting. Provided the authors are willing to revise the manuscript according to reviewers' suggestions, we will reconsider it again.

We would appreciate receiving your revised manuscript by Mar 29 2020 11:59PM. To enhance the reproducibility of your results, we recommend that if applicable you deposit your laboratory protocols in protocols.io, where a protocol can be assigned its own identifier (DOI) such that it can be cited independently in the future. For instructions see: http://journals.plos.org/plosone/s/submission-guidelines#loc-laboratory-protocols

We look forward to receiving your revised manuscript.

Kind regards,

Prof. Raffaele Serra, M.D., Ph.D

Academic Editor

PLOS ONE

Additional Editor Comments (if provided):

The manuscript is potentially interesting. The authors must provide a new version of the manuscript taking into account the reviewers' comments.

Journal Requirements:

2. To comply with PLOS ONE submission requirements, in your Methods section, please provide additional information regarding the experiments involving animals and ensure you have included details on (1) methods of sacrifice, (2) methods of anesthesia and/or analgesia, and (3) efforts to alleviate suffering. Please also provide details of animal welfare (e.g., shelter, food, water, environmental enrichment) and the frequency with which animals were monitored for signs of suffering or distress.

- In your Methods section, please provide the full name of the animal ethics committee that approved the study protocol, as well as the approval or permit number that was issued upon approval.

3. Please include your tables as part of your main manuscript and remove the individual files. Please note that supplementary tables should remain uploaded as separate "supporting information" files

Reviewers' comments:

Reviewer's Responses to Questions

**Comments to the Author**

1. Is the manuscript technically sound, and do the data support the conclusions?

Reviewer #1: No

2. Has the statistical analysis been performed appropriately and rigorously? 

Reviewer #1: No

3. Have the authors made all data underlying the findings in their manuscript fully available?

Reviewer #1: No

4. Is the manuscript presented in an intelligible fashion and written in standard English?

Reviewer #1: Yes

5. Review Comments to the Author

Reviewer #1: In this paper, the authors describe results from a rat lymphedema model in which lymphatic vessels and nodes are removed and the resulting limb volume changes and changes within the skin are described. The observed pathology (namely fibrosis and adipocyte expansion), the methods for quantifying changes in function with ICG, or the animal model itself are all variations of work reported by others. In the authors attempt to establish the novelty of the findings, many of the claims are grossly overstated and overlook or over-simplify a significant body of literature regarding mechanisms of lymphedema development in animal models. In addition, the manuscript is missing the appropriate statistical analysis to draw conclusions. The authors need to clearly focus on the limitations that this model addresses, the novelty of the findings themselves, and the value the model adds to the field. Specifics of these are as follows:

1) Lipid accumulation and fibrosis by collagen deposition in the skin, while certainly necessary components of pathology in lymphedema models, has been reported in many other animal models (some of which have been cited) and some which are missing: (Swartz, J Biomech, 1999; Mendez, AJP, 2012; Savetsky, AJP, 2014). The authors claim “Moreover, for the first-time SL stage migration was validated by characteristic histological alterations, e.g. subdermal mast cell infiltration, adipose tissue deposition, and fibrosis by increased skin collagen expression.” While the term stage migration is not used in the non-clinical lymphedema literature (including reference 12 which the authors cite on line 54 to establish the term), many of the papers by the research groups of Melody Swartz, Babak Mehrara, Jeremy Goldman, Brandon Dixon, Michael Detmar, Stan Rockson, etc.. report events that describe the key features of this process in animal models. It is an overstatement to suggest the observations in this model have not been made before. Another example “Stage migration is a key feature of the disease but, upon now, it has never been reported in an animal model as a consistent process.” (line 87). In their effort to oversell the model, the importance of the novelty of the model and what it could be used for is lost.

2) In addition, animal models of lymphedema have been crucial in understanding many of the molecular mechanisms important in lymphedema development, a few of which have been cited but many which are missing: (Nores, Nat Comm, 2018; Nores, J Inv Det, 2017; Tian, Sci Trans Med, 2017; Ogata, J Inv Det, 2016; Gousopoulos, J Inv Det, 2017) which have also provided therapeutic targets that have been effective in the animal models, albeit their clinical effectiveness remains to be established (Gardenier, Nat Comm, 2017; Tian, Sci Trans Med, 2017; Honkonen, Ann Surg, 2013). The paper here provides no new mechanistic insight or evidence that the model can address the shortcomings of these other models in yielding new mechanistic insight and statements like “and the molecular key players of stage migration have not been yet identified” (line 63) are incorrect and give the impression that the authors are unaware of the work in the field that sets the context for their animal model.

3) NIR imaging has been used both to access lymphatic function in lymphedema models (Gousopoulos, Am J Path, 2016; Kwon, PLoS One, 2014; Nores, Nat Comm, 2018) and to guide lymphatic injury in creating animal models of lymphedema (Weiler, Sci Rep, 2019). The authors claim on line 79 “… yet it has not been applied so far for in-vivo studies” is incorrect.

4) The statement in the abstract “In contrast to the few models reported so far, in this study the authors decidedly avoided the use of radiation for the lymphedema induction” is a significant overstatement and generalization of the lymphedema literature given all of the same papers mentioned above. In addition, many animal models of lymphedema have been created without radiation (above papers), and have shown to be made more severe with radiation (Lynch, AJP, 2015; Avraham, AJP, 2010).

5) In the statistical analysis section the authors state that they used used a one-way ANOVA. While this tells you that the means are different, it does not make any claims as to which means are different from one another. Also it is not clear from the figures, when any of these tests were actually applied and what the results of the test are. If making multiple comparisons (as should be done in Fig 2, 3, and 5) a a post-hoc correction must be applied. The same is true for the Wilcoxon signed-rank test. Furthermore, since the appropriate statistical test is often different for each figure, the test used to determine significance for each figure should be clearly stated.

6) The figures are very poor resolution and some of the labels and fonts are difficult to read.

7) Measuring fluorescence at a single time point to quantify lymphatic function could produce erroneous results as the clearance of tracer by lymphatics has been shown to be exponential (Karlsen, AJP, 2012; Karlsen, ATCB, 2018). In addition, repeat injections of ICG has been shown to be problematic in inducing an inflammatory response (Weiler, Front Phys, 2013) additionally raw ICG fluorescence is highly dependent on protein binding (Weiler, J Biom Opt, 2012) and thus differences in fluorescent intensity (Fig 3) are difficult to interpret.

8) The water displacement technique seems like it could be highly subjective to error as the method drastically depends on how far one submerges the limb into the container. The authors have done their best to control for this but it would be helpful to have a few measurements of intraoperator and interoperator variability of a few blinded users to get a better appreciation of how accurate and repeatable the measurement is.

9) Normalizing the life span of a rat and a human to equate the temporal kinetics of lymphedema development (i.e. that 1 year equals 11.8 days) is not appropriate. The remodeling of tissue depends on the turnover rates of matrix constituents, the proliferation rates of cells, etc… and these do not scale linearly with life span. In addition the statement that “nearly all reported incidences of SL were reported in the first 3 years” is not correct. While most large clinical studies have stopped tracking patients after 36 months, those that have followed patients further out continue to observe new occurrences of lymphedema long after 36 months (Armer, Lymphology, 2010; Petrek, Cancer, 2001).

10) One of the papers’ points is that the tail model damages more than just the lymphatics and thus is not clinically equivalent to human lymphedema. Given that a large amount of our understanding of the molecular mechanism’s involved in lymphedema are from the mouse tail model, there would be value in alternative models that overcome its limitations in order to understand new biology. However, it appears that all of the results in the pathology in this surgical model in the rat are identical to what has been shown in the mouse tail – lipid accumulation, mast cell infiltration, fibrosis and collagen deposition, impaired ICG clearance, etc… It would be helpful if the authors could emphasize the data in this model that produces unique results over models whose limitation it is seeking to overcome.

6. PLOS authors have the option to publish the peer review history of their article (what does this mean?). If published, this will include your full peer review and any attached files.

Reviewer #1: No

---

## [Author Response · Author response to Decision Letter 0]

25 Mar 2020

A detailed response to all the reviewers comments and concerns is provided in an attached file named "response to reviewers_plosOne". We are thankful for the very helpful literature suggestions and proposed corrections, provided by the reviewers.

---

## [Decision Letter · Decision Letter 1]

7 May 2020

PONE-D-20-00067R1

Evidence of stage migration in a novel, validated fluorescence-navigated and microsurgical-assisted secondary lymphedema rodent model

PLOS ONE

Dear Mr. Will,

Thank you for submitting your manuscript to PLOS ONE. After careful consideration, we feel that it has merit but does not fully meet PLOS ONE’s publication criteria as it currently stands. Therefore, we invite you to submit a revised version of the manuscript that addresses the points raised during the review process.

The manuscript was improved but there are still some concerns to be addressed by reviewer #1. Please amend your manuscript accordingly.

We would appreciate receiving your revised manuscript by Jun 21 2020 11:59PM. To enhance the reproducibility of your results, we recommend that if applicable you deposit your laboratory protocols in protocols.io, where a protocol can be assigned its own identifier (DOI) such that it can be cited independently in the future. For instructions see: http://journals.plos.org/plosone/s/submission-guidelines#loc-laboratory-protocols

We look forward to receiving your revised manuscript.

Kind regards,

Prof. Raffaele Serra, M.D., Ph.D

Academic Editor

PLOS ONE

Additional Editor Comments (if provided):

The manuscript was improved but there are still some major revisions that need to be made. Please see reviewers' comments.

Reviewers' comments:

Reviewer's Responses to Questions

**Comments to the Author**

1. If the authors have adequately addressed your comments raised in a previous round of review and you feel that this manuscript is now acceptable for publication, you may indicate that here to bypass the “Comments to the Author” section, enter your conflict of interest statement in the “Confidential to Editor” section, and submit your "Accept" recommendation.

Reviewer #1: (No Response)

Reviewer #2: All comments have been addressed

2. Is the manuscript technically sound, and do the data support the conclusions?

Reviewer #1: Partly

Reviewer #2: Yes

3. Has the statistical analysis been performed appropriately and rigorously? 

Reviewer #1: Yes

Reviewer #2: Yes

4. Have the authors made all data underlying the findings in their manuscript fully available?

Reviewer #1: No

Reviewer #2: Yes

5. Is the manuscript presented in an intelligible fashion and written in standard English?

Reviewer #1: No

Reviewer #2: Yes

6. Review Comments to the Author

Reviewer #1: In this revised manuscript the authors have made some strong improvements including a detailed statistical methods section with the appropriate statistics applied and a broader inclusion of the relevant scientific literature to the into and discussion. There still are some major revisions that need to be made, some of which were brought up in the previous review and were not adequately addressed.

Major comments

1. I do not think it can be determined what exactly is being measured with the NIR imaging approach the way it was conducted here. As mentioned previously, when ICG is injected it binds to protein and there is a shift in the excitation and emission spectra and a gain in the quantum yield which has been shown to result in an increase in fluorescence of up to 500%. I would hypothesize that the total fluorescence intensity of an image in one of the operated rats taken at 1 hour after injection would be extremely brighter than an image taken immediately after injection. This is because the dye binds to protein before it is cleared by lymphatics, the intensity goes way up, and then the brighter ICG carried away where it leaks out further downstream. There was a lot of work done here and I am not asking that the NIR clearance studies be redone. However I do not think you can make definitive statements about these measurements relating quantitively to the clearance capacity of the lymphatics.

2. It is still not clear how the NIR images were quantified to produce the plots from the description in the methods: “Furthermore, the ICG fluorescence signal at one-hour post application of the gated extremities were gathered as counts per second (cps) by the VIS-NIR X-ray imaging optical device.” Usually in the context of fluorescence the term gating refers to setting a threshold for the intensity, below which the signal is considered 0. I think what the authors mean here is that they selected a ROI to include just the extremities and then calculated the fluorescence here. The authors also mentioned they validated the technique by imaging continuously for 1 hour in unoperated rats. I think this data would be helpful to include to provide a sense of how much dye is actually being cleared during this 1 hour time period. The statement that it is almost all gone contradicts several published reports that ICG can stick around for days at the site of injection.

3. In addition to a new approach to generating a lymphedema model in the rat, the primary new finding of the manuscript is the presence of this aSMA+/Vimentin+/HSP+ cell type that is present in the skin at later stages. It would add tremendous value to this data if the authors could figure out a way to better capture this rich data set across the different time points and locations, similar as to what was done for Figure 5 for collagen. Could the number of aSMA+/Vimentin+/HSP cells be counted to show their increase over disease progression? Is this localized primarily to the location of the initial lymphatic injury or is it seen in the distal end of the injured limb?

4. The manuscript would be greatly enhanced by a figure that summarizes the kinetics of the pathological changes in this animal model. A primary strength of the model is the thoroughness with which the authors looked at lymphatic physiology, collagen, lipid, mast cells, and myofibroblasts at all of these different time-points in both the injured limb and the contralateral control. For the histology, this was done in three different zones as well

Minor comments

1. I recommend replacing the phrase “stage migration” with the more descriptive phrase “lymphedema stage progression”. The term “stage migration” has never been used in the lymphedema literature and it is not immediately obvious what exactly this term means. I assume that the authors are using this to refer to the advancement of the pathology through the various clinical lymphedema stages according to the classification stages used by the ISL. While they do define more clearly what they mean by the term on p.9, the term appear in the title, the abstract, and in numerous locations in the early pages of the paper before the reader is told what the term means.

2. “Unreliable in vivo generation of SL” (p. 3), while listed as a weakness by the authors of current models, could just as easily be considered a strength. The authors heavily emphasize the “translatability” of their model as it relates to human lymphedema. Wouldn’t the more translatable model be one were lymphedema incidence is not 100%, as it is nowhere close to this rate in the clinical surgeries associated with lymphedema incidence.

3. In the abstract I suggest replacing the phrase “translatable tool” with “clinically relevant model” as an model is not translatable in the general use of the work, which is typically reserved for therapies, technologies, etc.. that could be used in the clinic.

4. Remove the term “solid” from the subtitle at the top of page 13 as this adjective is vague in this context and does not add meaning.

5. Statistical markings (e.g. * and **) need to be directly on the figures so that the reader can directly determine which comparisons are significant and what the p value range is.

6. I think it is an overgeneralization to state there is a lack of surgical refinement and intraoperative mapping inherent to the rodent tail (p. 18). While I am sure there is a large variation across the different groups that have used this technique, collecting lymphatics can be easily visualized with methylene blue injection and carefully injured in a manner that avoids unspecific venous insufficiency (for example see the methods section describing the method in Tabibiazaret et al., PLoS Medicine, 2006).

7. Intro, page 3: Identification of lymphatics with dyes such as methylene blue and evan’s blue suffer the same limitations in regard to their lymphatic specificity as does ICG. When injected in small doses into the skin, they have a very high affinity to albumin, which they bind to and are then drained by lymphatics due to the molecular weight of albumin excluding their uptake by the circulation. ICG certainly enhances visualization of lymphatics if one has the appropriate optics, but a trained researcher can easily identify and remove lymphatics with blue dye labeling as well.

8. The next revision need to be carefully read for grammar as there are numerous mistakes throughout the manuscript.

9. This was mentioned in the last revision, but NIR imaging has been used to identify lymphatics and guide surgical injury in animal models in other publications: Weiler, Sci Rep, 2019 and Nelson, Nature Biom Eng, 2019. There is still a statement in the manuscript that NIR has nor been used to “guide a precise lymphatic injury for in vivo studies” (p.3) that is not accurate.

10. The authors need to replace references to rat tail in the manuscript with mouse tail as almost all lymphedema models in the tail have been done in mice. There are no good rat tail models of lymphedema due to the large amount of cartilage (as mentioned by the author), however this is not true of the mouse tail and the histological changes in the mouse tail are remarkably similar to the human and to those reported in the paper here.

11. I suggest the authors downplay how translational their model is. As mentioned before every animal undergoing surgery swells above a “lymphedema threshold”, the appearance of lymphedema occurs on the first day after surgery, and every animal’s swelling appears to be resolving on its own without any intervention. None of these three features are common in clinical lymphedema. This model is definitely valuable, but it seems to suffer the most crucial limitations of all current rodent models of lymphedema.

Reviewer #2: The authors have satisfactorily addressed the comments by the previous reviewers

7. PLOS authors have the option to publish the peer review history of their article (what does this mean?). If published, this will include your full peer review and any attached files.

Reviewer #1: No

Reviewer #2: No

---

## [Author Response · Author response to Decision Letter 1]

9 Jun 2020

Please see the specific comments for Reviewer#1 and Reviewer#2 in the uploaded "Response to reviewer II" as well as the previous rebuttal letter.

---

## [Decision Letter · Decision Letter 2]

26 Jun 2020

Evidence of stage progression in a novel, validated fluorescence-navigated and microsurgical-assisted secondary lymphedema rodent model

PONE-D-20-00067R2

Dear Dr. Will,

We’re pleased to inform you that your manuscript has been judged scientifically suitable for publication and will be formally accepted for publication once it meets all outstanding technical requirements.

Kind regards,

Prof. Raffaele Serra, M.D., Ph.D

Academic Editor

PLOS ONE

Additional Editor Comments (optional):

amended manuscript is acceptable

Reviewers' comments:

Reviewer's Responses to Questions

**Comments to the Author**

1. If the authors have adequately addressed your comments raised in a previous round of review and you feel that this manuscript is now acceptable for publication, you may indicate that here to bypass the “Comments to the Author” section, enter your conflict of interest statement in the “Confidential to Editor” section, and submit your "Accept" recommendation.

Reviewer #1: All comments have been addressed

2. Is the manuscript technically sound, and do the data support the conclusions?

Reviewer #1: Yes

3. Has the statistical analysis been performed appropriately and rigorously? 

Reviewer #1: Yes

4. Have the authors made all data underlying the findings in their manuscript fully available?

Reviewer #1: Yes

5. Is the manuscript presented in an intelligible fashion and written in standard English?

Reviewer #1: Yes

6. Review Comments to the Author

Reviewer #1: (No Response)

7. PLOS authors have the option to publish the peer review history of their article (what does this mean?). If published, this will include your full peer review and any attached files.

Reviewer #1: No

---

## [Editor Report · Acceptance letter]

8 Jul 2020

PONE-D-20-00067R2 

Evidence of stage progression in a novel, validated fluorescence-navigated and microsurgical-assisted secondary lymphedema rodent model 

Dear Dr. Will:

I'm pleased to inform you that your manuscript has been deemed suitable for publication in PLOS ONE. Congratulations! Your manuscript is now with our production department. 

Kind regards, 

on behalf of

Prof. Raffaele Serra 

Academic Editor

PLOS ONE